# Does the National Fitness Policy Promote National Health?—An Empirical Study from China

**DOI:** 10.3390/ijerph19159191

**Published:** 2022-07-27

**Authors:** Jun-Yi Zheng, Li-Xia Luan, Mei Sun

**Affiliations:** 1School of Physical Education, Huazhong University of Science & Technology, Wuhan 430074, China; jillz@hust.edu.cn (J.-Y.Z.); luanlixia@hust.edu.cn (L.-X.L.); 2College of Public Administration, Huazhong University of Science & Technology, Wuhan 430074, China

**Keywords:** national fitness policy, national health, human capital, public finance health expenditures, regional differences

## Abstract

The influence of national health level in the stability and sustainable development of national society is increasingly prominent. The purpose of this study is to examine whether, when, and how national fitness policies exert influence on national health. Panel data from 2008 to 2017 of 30 Chinese provinces (cities) (except the Tibet autonomous region) were used to systematically reveal the direct impact of national fitness policies on national health and its characteristics in different regions, as well as the interaction mechanisms of human capital and finance health expenditures in public sports. This study found that first, national fitness policies had a positive effect on adult health. Second, sports human capital weakens the health effect of national fitness policies, while public finance health expenditures strengthen this effect. Lastly, the health effect of national fitness policies varies significantly across regions due to uneven regional economic development, and the differences in the effects on different age groups (adults and children) are equally pronounced. This study suggests that national fitness public service system and diverse national fitness plans improving national health level are important for a new dynamic balance and high quality coordinated development in both Chinese economic growth and social welfare.

## 1. Introduction

Health is the most significant human capital and is an important means to promote sustainable socioeconomic development [1]. However, in the context of the COVID-19 pandemic and traditional social problems such as aging population and chronic diseases in the information age, public health faces escalating challenges. As a result, governments have incorporated improving national health into Sustainable Development Goals and has gradually gained prominence in global and national health policies. It effectively increases the life expectancy of the population and promote public health well-being and equity [2]. The most common non-medical intervention adopted by governments involves encouraging people to actively participate in physical fitness activities for the purpose of disease prevention and health promotion [3,4].

The Chinese government has always provided great importance to national health and considered it a people-oriented issue. Promoting national fitness activities is an important part of the “National Strategy for a Healthy China”. In the fight against the COVID-19 pandemic, the Chinese government has emphasized that “the safety and health of the people should come first” [5], which highlights the importance of national health. At present, China has built a relatively complete medical and health care system, but given the population base, economic base, regional development, and other social problems, the overall public health supply is insufficient, and the health service providers are relatively fewer; hence, the national health needs cannot be guaranteed in an all-round and full life cycle. Furthermore, the rapid urbanization brought by China’s socioeconomic transformation has further intensified the conflicts between public health and social equality.

As a prerequisite for health practice, formulation, promulgation, and implementation of a national fitness policy are positive responses to the public service system of national fitness and macroeconomic regulation of health governance. In addition, it is an important initiative to meet national health needs and promote sustainable national economic development. Research in the area of public services for national health can enable policy-makers to optimize investments, promote policy foresight and rationality, and thereby achieve greater socioeconomic benefits [6]. Although existing studies have noted the positive association of national health promotion with fitness policies, they have not yet clarified their utility value and the pathways and mechanisms of their impact on people’s well-being. Most studies have focused on environmental science, health economics, epidemiology, and demography [7,8]. Scholars have paid more attention to the group-differentiated characteristics of environmental pollution [9,10], health investments and returns in the human capital perspective, the mechanisms of different diseases [11,12], and health inequalities arising from social structures and institutions [13]. In fact, for public health services, the above-mentioned studies focus too much on micro-individual and group studies that intermingle environmental, organizational, and group distractions, ignoring the fact that mass sport behavior as a social behavior is more related to the influence of public health inputs and outputs at the national level. In general, positive national health outcomes are more likely to be attributed to health policy factors that create favorable conditions for the optimization of internal mechanisms and the improvement of the external environment. However, there is a lack of literature on this topic, which is not conducive to pioneering academic inquiry and future government practice.

Based on the literature review and analysis, this study concludes that there are still some issues that need further consideration in the study of the influencing factors on national health. First, most previous literature have used a linear framework to determine the direct relationship between environmental pollution, economic development, and public health [14,15,16]. A few studies have involved quantitative analysis of the causal relationship between public sports policies and health, and even fewer studies have explored the non-linear moderating effects. Second, most of the studies based on Chinese samples focus on the overall analysis of Chinese provinces and autonomous regions or a few key cities, ignoring the rapid but unbalanced economic development among regions; hence, an in-depth exploration of health policy effects among different regions in the overall national sample is necessary. It is important to consider the effects of the national fitness public service system, its policy effectiveness in promoting national health development, and its possible influential mechanisms, especially to increase the research on the differences in the health effects between regions.

This study differs from previous intuitive analyses in that instead of looking into environmental, health, medical, and economic perspectives of national health, it takes a cross-disciplinary approach, such as public administration and physical education. To our knowledge, this is the first study to use cross-provincial (municipal) level panel data in China from 2008–2017, which were validated by constructing an empirical model to deeply reveal the intrinsic mechanisms and paths of national fitness policies affecting national health levels; the resulting problem of equilibrium in the supply of public services for national fitness were discussed. The main contributions of this study are: first, based on the panel data of 30 provinces and autonomous regions (except Tibet) in China in the last decade, the effect of national fitness policy on national health is examined from a macro perspective, which provides a valuable and evidence-based reference for the development of the national strategy of “Healthy China”. Second, the empirical analysis using individual fixed-effects models can help overcome the shortcomings of the traditional linear regression method to examine the effects of explanatory variables on the conditional expectations of the explanatory variables, and can reveal the effects of national fitness policies on national health in a comprehensive and stable manner. Third, the relationship between the national fitness policy and national health is discussed by region, making the research findings more consistent with the reality of regional unbalanced development in China. In addition, our study involves the relationship between policy intensity, mortality, human capital, and financial investment, reflecting from the side that national health is not only influenced by the environment but also by other related factors. This helps to better understand the related occurrence mechanism and provide empirical support, and also provides positive inspirational effects and important references for existing theoretical research and practical exploration.

The succeeding sections reviews the evolution of national fitness policies in China and the progress of domestic and international research; introduces the data, variables, and model construction of the study; analyzes and discusses the data results; and highlights the main findings of the study and recommendations for future research. We focus on the determinants of the national fitness policy intensity change that may promote or hinder the national health level and their ways of action.

## 2. Practical Development and Theoretical Basis of China’s National Fitness Policy

### 2.1. The Evolution of China’s National Fitness Policy in Practice

China’s national fitness policy is an important institutional guarantee for improving national health and health literacy. The success of the 2008 Beijing Olympic Games shifted the public sports hotspot from competitive sports to national fitness, unveiling the prelude to a new era of comprehensive promotion of national fitness programs. Subsequently, national fitness and “Healthy China” became national strategies, and the government successively issued national fitness policies, such as the Regulations on National Fitness (State Decree No. 560), Guidance on Accelerating National Fitness into Families, and National Fitness Plan (2011–2015), National Fitness Plan (2016–2020), to construct a comprehensive range of basic public sports services. In particular, at the local government level, provinces in China have issued various forms of provincial Regulations on National Fitness (Promotion) in response to national policies, and conducted extensive national fitness sports activities. This study focuses on the national fitness policies in Chinese provinces during the booming period of national fitness since 2008, and analyzes the evolution of the intensity and content of national fitness policies from 2008 to 2017.

This paper mainly focuses on following types of national fitness policies from 2008 to 2017 in Chinese provinces: national fitness regulations which refers to those basic legal documents that set the direction of reform and development, service system and operation mechanism for national fitness, often with a certain strategic and holistic nature, such as sports law, national fitness regulations, regulations on public cultural and sports facilities, etc.; national fitness regulatory documents, which refers to guiding documents involving the design of the national fitness policy process and the implementation of responsibilities, they are the refinement of the national fitness regulations, such as the national fitness implementation plan, the implementation opinions on promoting the high-quality development of national fitness, the guidelines on the operation and maintenance management of national fitness venues and facilities, etc.; relevant national fitness work documents referring to the supporting specific national fitness work guidance, such as the road running sports events and activities safety management guide, notice on further strengthening the supervision and management of outdoor sports events and activities, the mass sports competition and the National Fitness Games competition regulations, etc.; and a small number of administrative license approvals and local government regulations involving national fitness, such as the Provincial Sports Bureau’s notice on the main points of the province’s mass sports work, etc.

First, we discuss the evolution and development of national fitness policies in Chinese provinces. China’s national fitness policy system is a typical centrally led paradigm, with local governments in each province developing national fitness policies and refining rules in response to the central policies, and providing public sports services as well as scientific fitness guidance directly to the public [17]. In the manuscript, initial intensity of national fitness policies in China in 2008 and national fitness policy intensity and policy intensity accumulation of provinces in China in 2017 are national fitness policies in China for a specific year interval. These numbers are authentic since they are based on Chinese practice. In terms of Figure 1, more than half of the other provinces have not yet issued supporting national fitness regulations, so the minimum value is specified as 0. Thus, starting from 0, with 1 as the unit of measurement, each increase of one unit indicates that the true increment of the number of national fitness policies is 1. The largest number of initial policies is in Zhejiang province, with a policy intensity of 7. To facilitate comparison of the effects, the scale interval is therefore set to 0–8, i.e., the true number of initial intensity of national fitness policies in China in 2008 is 0–8. The intensity of national fitness policies in each province in 2008 shows that the introduction of national fitness policies in China is in the initial stage. Only a total of 14 provinces, including Zhejiang, Jiangsu, Anhui, Fujian, and Inner Mongolia, achieved zero breakthrough in national fitness policies. This result indicates that the construction and development of relevant policies in Chinese provinces are obviously insufficient. The 14 provinces that have issued national fitness policies and regulations include both economically developed and coastal provinces in the east such as Shanghai, Zhejiang, and Fujian province, which have a high level of health literacy among residents, as well as underdeveloped regions in the central and western parts of China that have quite ethnic characteristics and sufficient natural resources such as Chongqing, Inner Mongolia, and Yunnan province. In addition, the starting intensity of the national fitness policy in 2008 in economically developed regions such as Beijing, Tianjin and Guangdong and underdeveloped regions such as Liaoning, Qinghai and Gansu were all zero, which did not show full consistency with regional economic development or resource distribution, so their specific mechanisms of action and effects need to be further explored.

This study plots the national fitness policies in each province in 2017 and the cumulative intensity of national fitness policies from 2008 to 2017 to present the changes in the intensity of national fitness policies in each province of China visually. As shown in Figure 2, there is a significant increase in the intensity of national fitness policies by provinces in China in 2017 compared to 2008. Same as the above, Figure 2 shows that national fitness policy intensity in China in 2017 involves a real policy quantity interval of 0–200, starting from 0, with 200 as the unit of measurement, and each increase of one unit indicates that the real increment of the number of national fitness policies is 200. Among them, there are still some provinces where the specification of the number of national fitness policies tends to be 0, such as Tianjin; Fujian province has the highest number of 200 policies. The real policy quantity interval of policy intensity accumulation of provinces in China from 2008 to 2017 starts from 0, with the lowest in Tianjin, where the number of policy accumulation is 0; the highest in Fujian, where the number of policies is 1084. After several debugging of the scale, it is found that contrast effect is the most obvious in the scale of 200. In addition, with a scale of 200, the number of provinces in the interval above 200 and below 200 is basically the same. Therefore, the true interval for the number of policy intensity accumulation of provinces in China from 2008 to 2017 is 0–1200. Overall, all provincial governments in China have basically achieved the coverage of national fitness policies. The analysis shows that the cumulative value of national fitness policy intensity exceeds 200 in more than half of the provinces, among which Fujian Province and Jiangsu Province even reach more than 800, and the construction of national fitness policies and related regulations has significantly accelerated. In addition, the unbalanced characteristics of national fitness policies among different regions are highlighted. For instance, the cumulative intensity of national fitness policies in Hainan Province in 2017 was 67, while the cumulative intensity of national fitness policies in Fujian Province in 2017 reached 1084. The reason for this is that although both the Hainan and Fujian Provinces are coastal cities, there are significant differences in their population density, economic development level, natural conditions, and resource carrying capacity. The specific mechanism and effect of the influence of different factors on the intensity difference of national fitness policies between regions need to be further explored.

Second, in terms of policy content, the focus of national fitness policy has shifted from the layout of infrastructure such as venue facilities, organization construction, fitness activities, scientific guidance, and physical fitness monitoring to the development and optimization of the public service system of national fitness. In August 2009, the Regulations on National Fitness was the first special regulation on national fitness in China which clarified the right of citizens to participate in national fitness and its legal status. National fitness became an essential livelihood project for national social development. Subsequently, central and local government policies on various elements of national fitness, such as social sports instructors, sports facility construction, fitness activities, fitness environment, social capital, and supervision services, have been continuously introduced to drive the construction of national fitness public service systems in all aspects [18]. For example, in 2011, the General Administration of Sports issued the Social Sports Instructor Development Plan (2011–2015), which clearly regulates the construction of social sports instructors. In 2015, Jiangsu Province issued the Implementation Opinions on Accelerating the Development of Sports Industry and Promoting Sports Consumption, which mobilizes social capital to strengthen the Jiangsu practice of national fitness, and takes the lead in the country to build a public sports service system demonstration area with clear functions, a sound network, and benefits for all.

Relying on the characteristic advantages of regional resources, several characteristic national fitness projects have emerged in some provinces and localities, such as the Guangdong Greenway Sports Project, Fujian Rainbow Trail Project, the Colorful Yunnan National Fitness Project, and the National Fitness Corridor along the Wei River in Shaanxi. The National Fitness Plan (2016–2020) proposes to deepen sports reform and create a new fashion of national fitness, and the coverage of national fitness policies has gradually achieved precise positioning, developing from universal policies to focus on compensation governance for disadvantaged groups. Considering the differences in health status and resource allocation of people at different levels, the sports rights and participation needs of key groups represented by women, people with disabilities, the elderly, farmers, and ethnic minorities are more urgent [19]. The national fitness policy has changed from a non-equal and biased supply under stratified segmentation to an integrated and coordinated development; the central and local governments have gradually established an institutional balance mechanism for resource allocation in the supply of public services for national fitness by increasing policy efforts, strengthening health promotion, and optimizing resource matching.

This study concludes that the differences in the number of national fitness policies in different regions of China during 2008–2017 mainly stem from the level of regional economic development (based on GDP per capita) and health literacy of its residents. Take the 2017 GDP per capita ranking of Chinese provinces and cities as an example, the top 10 regions in descending order are Beijing, Shanghai, Tianjin, Jiangsu, Zhejiang, Fujian, Guangdong, Shandong, Inner Mongolia, and Chongqing; the bottom 10 in descending order are Sichuan, Qinghai, Jiangxi, Anhui, Shanxi, Heilongjiang, Guangxi, Guizhou, Yunnan, and Gansu. Overall, the number of national fitness policies in economically developed coastal areas (such as Zhejiang, Fujian, etc.) is significantly higher than that in less developed areas (such as Hunan, Hainan, Ningxia, Heilongjiang, etc.). On this basis, with the conditions of economic development, some developed regions (such as Jiangsu, Guangdong, etc.) have made full use of the advantages of regional resources to strengthen the policy inclination of national fitness projects, and are committed to building a sound public service system of national fitness. At the same time, we notice that there are also some special provinces (such as Sichuan, Anhui, Jiangxi, Gansu, etc.) that still maintain relatively high accumulation of national fitness policies despite the lack of economic development level. They are aided by the advantages of locations, ethnic characteristics, natural resources and environmental carrying capacity, and also their attempts to attract talents and investments by improving residents’ health environment and optimizing public service. We believe that the future of fitness policy will be a good one. Therefore, we believe that future research needs deep analysis on the specific elements of the differences in the number of national fitness policies in different regions through field research, expert interviews, and panel data to further supplement and improve existing research.

In summary, China’s national fitness policies have achieved leapfrog development during the period 2008–2017. National fitness and national health have gradually achieved deep integration, and public sports services have entered an active construction phase. Furthermore, the national fitness policy intensity in Chinese provinces increased significantly during this period, and policy-led national fitness group activities are widely conducted; research has shown that during this period, public participation is high and individual health literacy is improved [20]. However, the analysis and evaluation of the effects of national fitness policies have been mainly based on objective allocation quantities such as sports facilities, venues, and funding scales [17], and these relatively one-sided evaluation approaches have neglected the substantive examination of national health-related impact indicators, especially the regional quantitative analysis under specific socioeconomic and lifestyle conditions.

### 2.2. Literature Review

Due to the high material living standards and better public awareness of sports and related education systems in developed countries such as Europe and the United States, as well as some medium-sized developed countries, sports are considered an important activity that governments must provide for their citizens, and the public is generally aware of the important role of national fitness in improving their health and quality of life. As early as the 1960s, national fitness became a major means to improving chronic diseases among occupational groups, relieve social medical pressure and maintain social stability in these countries. In 1985, the International Olympic Committee established the Mass Sports Commission to call for a World Congress of Mass Sports. In the following year, the first World Congress of Mass Sports was held in Cologne, Germany, to exchange experiences on the themes of practical and theoretical development of mass sports, stakeholder relations, health and social benefits, and future development, which further set off a worldwide boom in mass sports activities.

Since the 1990s, major western developed countries have promulgated national fitness and mass sports policies, and mass sports have basically achieved comprehensive popularity; the value of sports has been generally recognized and respected by the public in the integration of health values. The U.S. Healthy People Report is considered the beginning of the global Healthy People program, which is officially renamed as the National Fitness Program in 2010. The plan has a clear aim of “changing people’s behavior through sports”, and includes both universal national health promotion policies and specific prevention and control programs for some diseases (such as arthritis, heart disease, obesity, etc.), involving business and industry, education, health care, public fitness and recreation, urban transportation planning, volunteer organizations, and other fields. This emphasizes the transformation of basic health education, the professionalism of health institutions, and the operability of strategy implementation [21]. The British National Health Service system is characterized by high welfare and wide coverage. The government has introduced the National Health Insurance Act, the National Health Service Act, and the Health and Social Care Act to improve the health of the nation, establishing a unified framework and management system for the National Health Service. It has strengthened front-end preventive health care and community services through decentralization, entrusted purchase, quality control, and service integration, and guided the public to actively strengthen self-health management. The quality of Cuba’s Fitness System is among the highest in the world for its community and family centered approach and emphasis on a preventive approach to health. The Cuban model aims to bring health care back to the community through a continuous effort to maintaining quality of service via the collection of health statistics from primary health care providers serving families and communities, and the regular review of health care quality by public health officials. This model of community-based, neighborhood-based, collective social structure for universal health care also provides valuable lessons and references for other countries. Other countries such as Norway, Sweden, Australia, and Japan have established relatively well-developed national fitness programs and mass sports regulations, combining sports activities with their related environment, public health, education, and media campaigns to create feasible conditions to make sports an important part of the public’s daily life.

Based on the above management practices, research on trend prediction, group difference analysis, policy effect assessment and influencing factors, and policy process of national health have emerged from the academic community. For example, Porter et al. [22] systematically assessed the relationship between exercise behaviors (physical activity, sedentary time, screen time) and cardiovascular health in adolescents based on the National Fitness Program guidelines using data from the 2012 NHANES Adolescent Health Survey. Based on public fitness and exercise registry data from the Fitness Registry and the Importance of Exercise National Database (FRIEND Registry), Nevill et al. [23] used multiple regression and anisotropic growth models to derive an updated predictive model of public cardiorespiratory fitness improvement in the U.S. Rolfe [24] developed a simple assessment tool to examine the state of urban public sport investment in regional Queensland, Australia, focusing on the potential economic and social benefits of public sport investment in infrastructure and project delivery. Polyakova and Ramchandani [25] examined regular users’ perceptions of the quality of public sports centers in England and analyzed the possible challenges and obstacles of multifunctional public sports facilities in creating sustainable sports participation. Fernández-Martínez et al. [26] examined the satisfaction and loyalty of users of public sports and health services aged 12–16 years in Spain at different stages and the possible impact of different dimensions of service quality on them. Carrad et al. [27] used a multi-case study to explore the process of organizational change and its possible perceived facilitators and potential barriers in the implementation of a health promotion program in New South Wales, Australia, specifying that customized health promotion strategies in different contexts require the involvement of leaders and the organization of practical actions.

In summary, extant research focuses on two aspects: the overall assessment and analysis of the implementation process of national health promotion programs through case studies in different cultural contexts and social relationships, and the examination of the effectiveness of national fitness policy implementation on public health based on micro-individual health indicators. However, the direct health effects of national health policies at the regional level have not been sufficiently discussed. Based on this, this study examines the effects of national fitness programs on national health in China and its possible regional differences using panel data at a macro level. We try to reveal the direct correlation between national fitness policies and national health and explore the potential mechanisms that may influence national fitness policies to promote or inhibit national health.

## 3. Materials and Methods

### 3.1. Variable Selection

In order to explore the effect and mechanism of the influence of national fitness policy on national health through regression analysis, we conduct data selection and construct an econometric model. The data are obtained from the China Statistical Yearbook, China Health and Health Statistical Yearbook, China Environmental Yearbook, China Environmental Statistical Yearbook, some statistical yearbooks of Chinese provinces and cities, and the Legal Star database. The sample selection interval is 10 years from 2008 to 2017. Due to the lack of corresponding data in Tibet, the data from this region is excluded.

As far as the explanatory variables are concerned, this study uses adult mortality and child mortality to reflect the national health level. The mortality data in this study are obtained from a study by Zhou et al., (2019) [28], which is published in the prestigious medical journal The Lancet with high reliability, and applies methods in GBD (the Global Burden of Diseases, Injuries, and Risk Factors Study) to systematically analyze mortality, morbidity, and risk factors under the health model of provincial administrative units in China. The core national health surveillance indicators involved in the existing studies are diverse and complex, and no consensus has been reached. Scholars have used average life expectancy (ALE), neonatal mortality rate (NMR), infant mortality rate (IMR), maternal mortality ratio (MMR), and malaria incidence (MI) as important indicators to evaluate the health level of a given region based on regional differences and research purposes [29,30]. Based on the national population and the coverage of the population served, this study follows the study of Sun and Li (2017) [31] and takes population mortality rate to assess the national health level, aiming to explore the overall level of national health and possible regional differences in the local context of China. It has been established that child health and child mortality are important factors of health for all policies, according to Timmons et al., (2012) [4], Guo and Huang (2019) [5], and Zhou and Gao (2021) [18]. In the case of China, existing studies on health outcomes for all have mostly focused on adult health, neglecting to explore child health and child mortality. Currently, various forms of children’s physical development centers have emerged in China during the promotion of the national fitness project, which increases the focus on children’s health. The realistic development needs are consistent with the selection and setting of variables in this paper. Therefore, in order to analyze the health effects of national fitness policies on different groups, this study follows Zhou et al., (2017)’s [28] definition and further divides mortality rates into adult mortality (aged from 15 to 60) and child mortality (aged under 5). In general, the lower the adult mortality rate and child mortality rate, the higher the level of national health.

In terms of explanatory variables, this study uses the cumulative number of national fitness policies promulgated by each province in China over the years as an indicator of the national fitness policy intensity. The data are obtained from the Legal Star Database, which is the most authoritative, informative, and updated regulatory database in China, covering all kinds of existing laws, administrative regulations, departmental regulations, judicial interpretations, local regulations, rules, normative documents, etc., approved and promulgated by the central government and local governments in China, as well as the provincial governments’ official websites. We conduct a comprehensive search of national fitness policies and regulations from 2008 to 2017 by province according to keywords, which are national fitness, (national) fitness program, (mass) sports, public sports, universal health, mass (sports) participation. Considering the cumulative effect of policies [32], it is generally believed that the greater the number of policies introduced within a certain period, the stronger the willingness and governance ability of local governments to treat specific issues [33]; thus, the cumulative effect of policies may have an important impact on the research results [34]. Accordingly, this study takes the cumulative number of policies in a specific period as a measure of the national fitness policy intensity.

Meanwhile, this study also includes two explanatory variables in the model, namely, practitioners and financial expenditures, to explore the key mechanisms of the relationship between national fitness policy intensity and national health. In practice, there are two ways to evaluate the supply efficiency of public goods, one is to measure it by calculating the input-output ratio, and the other is to establish relatively quantifiable indicators, i.e., efficiency criteria [35]. Since the former output is difficult to quantify accurately, most of the existing literature evaluates and analyzes financial expenditures through evaluation indicators. This study draws on Liu’s [36] research on government sports investment indicators, and selects personnel and expenditures, i.e., the number of sports practitioners per 10,000 population and finance health expenditures per capita as indicators, which can better reflect the efficiency of public goods supply inputs and equilibrium outputs. Since information on China’s finance is not easily accessible, it is difficult to collect independent data on sports finance expenditures. Thus, this study uses culture, sports, and media finance expenditures recorded in statistical yearbooks instead. Since China’s reform and opening up, culture and media have become highly market-oriented and are much less dependent on the state funds than sports, so the vast majority of culture, sports, and media expenditures are still invested in sports, and it is reasonable to choose this expenditure to represent the state’s financial expenditures in sports.

Finally, as far as the control variables are concerned, nine factors such as social environment, health care, economy, and population, which are closely related to national health, were selected in this study. Specifically, they include: health care expenditures per capita (total health care expenditures/total population), health education institutes/stations (number of public health care institutions, such as infirmaries, nursing stations, etc.), PM_2.5_ (degree of air pollution), park green space per capita (area of urban green space/total population), urban sewage treatment rate (proportion of treated sewage to total sewage discharge), household waste harmless treatment rate (the proportion of harmlessly treated domestic waste to total waste intake), GDP per capita (gross domestic product per capita), urbanization rate (proportion of urban population), and population density (total population/land area).

Data on health care expenditures per capita and health education institutes/stations are obtained from the China Health Statistical Yearbook, data on PM_2.5_, sewage treatment rate, garbage treatment rate, and park green space per capita are obtained from the China Environment Yearbook and the China Environment Statistical Yearbook, and data on per capita GDP, population density, and urbanization rate are obtained from the China Statistical Yearbook. Numerous studies have confirmed the negative impact of environmental problems on national health, and the spillover of air pollution is stronger compared with other environmental factors, and this study uses annual PM_2.5_ emissions to measure the degree of air pollution [31]. Other environmental issues such as water pollution, waste disposal, and greening have also become important challenges that currently threaten people’s lives and health and may have a confounding effect on national health [37,38]. Studies have also found that personal economic income as well as health care expenditures could significantly affect changes in national health status [39,40]. Further, in a study on the efficiency of public sports services in 31 provincial governments in China, Li et al. [41] have found that two GDP per capita indicators (sports stadium area per capita and sports expenditure per capita) have a positive effect on the efficiency of public sports services, while population density and educational literacy are significantly related to the services’ efficiency. Hence, GDP per capita, population density, and educational literacy also become important factors that may influence national health. Moreover, in the context of intense urbanization and modernization, the health status of urban residents usually faces greater risks and challenges compared to rural residents.

### 3.2. Descriptive Statistics

The sample data in this study involves 30 provinces, cities, and autonomous regions in China except Tibet, which spans a 10-year period from 2008 to 2017, with a total of 300 samples. Table 1 reports the results of descriptive statistics and expected signs for each variable. The differences between the maximum and minimum values of culture, sports, and media practitioners per 10,000 population and financial expenditures on culture, sports, and media per capita are significant and may be important factors influencing the relationship between policy intensity and national health (adult mortality and child mortality).

### 3.3. Model Specifications

An econometric model is constructed in this study to analyze the effect of policy intensity on national health. Among them, $y_{it}$ represents the explained variable, i.e., adult mortality or child mortality; $Policy_{it}$ is the main explanatory variable in this study, representing the policy intensity in each province in a specific year; $X_{it}^{\prime }$ represents the above-mentioned control variables; $\alpha_{1}$ and β analysis represent the regression coefficients of the explanatory and control variables, and $\delta_{i}$ and $\epsilon_{it}$ represent the error terms.
(1)$$y_{it}=\alpha_{0}+\alpha_{1}Policy_{it}+\;X_{it}^{\prime }\beta +\gamma_{t}+\delta_{i}+\epsilon_{it}$$

In order to explore the specific mechanism of the effect of the policy intensity of national fitness on national health, this study constructs an econometric model with cross-sectional terms on the basis of the above econometric model. Among them, $Policy_{it}Employee_{it}$ is the interaction term between policy intensity and practitioners, and $Policy_{it}Expendit_{it}$ is the interaction term between policy intensity and financial expenditures. Through this model, we explore the interaction between the policy intensity on national fitness and the related practitioners and financial expenditures.
(2)$$y_{it}=\alpha_{0}+\alpha_{1}Policy_{it}+\alpha_{2}Policy_{it}Employee_{it}+\alpha_{3}Policy_{it}Expendit_{it}+X_{it}^{\prime }\beta +\gamma_{t}+\delta_{i}+\epsilon_{it}$$

Finally, considering that this study used multi-year panel data, based on the results of the Hausman test, we used an individual fixed-effects model in the regression analysis to effectively eliminate time-invariant factors.

## 4. Results and Analysis

### 4.1. Direct Effect of National Fitness Policy Intensity on National Health

Table 2 shows the influence of national fitness policy intensity on adult and child mortality. Among the control variables, health education institutes/stations (r = −0.000002, *p* < 0.01), and green space per capita (r = −0.000009, *p* < 0.01) are significantly negatively correlated with adult mortality, while other control variables are not. There is a significant negative correlation between the national fitness policy intensity and adult mortality (r = −0.000001, *p* < 0.01); the greater the national fitness policy intensity, the lower the adult mortality rate.

Among the control variables, health expenditure per capita (r = −0.002, *p* < 0.01), green space per capita (r = −0.003, *p* < 0.001), sewage treatment rate (r = −0.004, *p* < 0.001), urbanization rate (r = −0.018, *p* < 0.001) are significantly negatively correlated with child mortality. The larger the green space per capita, the higher the sewage treatment rate, the higher the urbanization rate, then the lower the child mortality rate. In addition, there is a significant positive correlation between population density and child mortality (r = 0.002, *p* < 0.001); the higher the population density, the higher the child mortality rate. There is also a significant positive correlation between national fitness policy intensity and child mortality rate (r = 0.0003, *p* < 0.01); the higher the national fitness policy intensity, the higher the child mortality rate.

### 4.2. Interactive Effects of Practitioners and Financial Expenditures

As shown in Table 3, there is a significant positive correlation (*p* < 0.01) between the interaction term between national fitness policy intensity and culture, sports, and media workers per 10,000 population and adult mortality. The main effect between policy intensity and adult mortality is significantly negatively correlated (*p* < 0.01), so there is a significant negative moderating effect of culture, sports, and media practitioners per 10,000 population between national fitness policy intensity and adult mortality. The more culture, sports, and media practitioners per 10,000 population, the weaker the relationship between national fitness policy intensity and adult mortality. Furthermore, there is a significant negative correlation (*p* < 0.05) between financial expenditures on culture, sports, and media per capita and adult mortality, i.e., the higher the financial expenditures, the lower the adult mortality. The interaction term between national fitness policy intensity and finance expenditures on culture, sports, and media per capita has a significant negative correlation (*p* < 0.01) with adult mortality. The main effect between policy intensity and adult mortality is significantly negatively correlated (*p* < 0.01), so there is a significant positive moderating effect of finance expenditures on culture, sports, and media per capita between national fitness policy intensity and adult mortality. The higher the finance expenditures on culture, sports, and media per capita, the stronger the relationship between national fitness policy intensity and adult mortality.

As shown in Table 3, there is a significant positive correlation between culture, sports, and media practitioners per 10,000 population and child mortality (r = 0.001, *p* < 0.05), but the interaction term between culture, sports, and media practitioners per 10,000 population and policy intensity has no significant correlation with child mortality (*p* > 0.1); the moderating effect of practitioners is not significant. There is a negative correlation between financial expenditures on culture, sports, and media per capita and child mortality, but its correlation is not significant (*p* > 0.1). Moreover, the interaction term between national fitness policy intensity and financial expenditures on culture, sports, and media per capita is also not significantly correlated with child mortality (*p* > 0.1), and the amount of financial expenditures has no significant effect on the relationship between national fitness policy intensity and child mortality.

### 4.3. Comparison of Regional Differences

In this study, we divide the 30 provinces, cities, and autonomous regions of China based on the GDP per capita ranking in 2017, and mark the top 15 as developed regions and the bottom 15 as underdeveloped regions. We first conducted a comparison of the effects of the relationship between national fitness policies intensity and adult mortality, and the results of the analysis are shown in Table 4. In developed regions, there is a significant negative correlation (*p* < 0.001) between national fitness policy intensity and adult mortality, i.e., the higher the policy intensity, the lower the adult mortality in developed regions. On the other hand, in underdeveloped regions, there is no significant correlation (*p* > 0.1) between national fitness policy intensity and adult mortality.

As shown in Table 5, there is a significant negative relationship between culture, sports, and media practitioners per 10,000 population and adult mortality in developed regions (*p* < 0.05), i.e., the more practitioners, the lower the adult mortality in developed regions. The interaction term between national fitness policy intensity and culture, sports, and media practitioners per 10,000 population and adult mortality rate in developed regions is significantly positively correlated (*p* < 0.001). The main effect between policy intensity and adult mortality is significantly negatively correlated (*p* < 0.01), so there is a significant negative moderating effect of culture, sports, and media practitioners per 10,000 population between national fitness policy intensity and adult mortality in developed regions. On the other hand, in underdeveloped regions there is a significant negative correlation between culture, sports, and media practitioners per 10,000 population and adult mortality (*p* < 0.05), i.e., the more practitioners, the lower the adult mortality in underdeveloped regions. However, as there is no correlation between policy intensity and adult mortality, and the interaction term of policy intensity and practitioners and adult mortality in underdeveloped areas is also not established (*p* > 0.1), thus culture, sports, and media practitioners per 10,000 population cannot significantly influence the relationship between them.

Table 5 also shows that there is a significant negative relationship between the interaction term of national fitness policy intensity and finance expenditures on culture, sports, and media per capita and adult mortality in developed regions (*p* < 0.001). The main effect between policy intensity and adult mortality is significantly negatively correlated (*p* < 0.01), so there is a significant positive moderating effect of finance expenditures between national fitness policy intensity and adult mortality in developed regions. On the other hand, in underdeveloped regions, there is a significant negative correlation between finance expenditures and adult mortality (*p* < 0.01), i.e., the higher the finance expenditures, the lower the adult mortality in underdeveloped regions. However, since there is no correlation between policy intensity and adult mortality and there is no significant correlation between the interaction term of finance expenditures and policy intensity and adult mortality in underdeveloped areas (*p* > 0.1), finance expenditures do not significantly affect the relationship between policy intensity and adult mortality in underdeveloped regions.

### 4.4. Heterogeneity Analysis of National Fitness Policy Intensity Affecting National Health

#### 4.4.1. Group Heterogeneity Analysis

The results show that national fitness policy intensity has a differential impact on adult mortality and child mortality. It may be due to the increasing financial expenditure on public sports, as well as the increasing amount of health education institutes/stations and park green space per capita. Chinese government departments at all levels provide great importance to improving the national health level and widely publicize and educate the public, which makes the public pay increased attention to their own health conditions. As a result, their willingness, number, and frequency of participating in sports and fitness activities continue to increase, thus promoting adult health. This can be interpreted as the gradual emergence of the “herd effect” of national fitness, resulting in a decline in adult mortality. However, this health effect has not positively affected children’s health. This study argues that this finding is a good clarification that the impact of established national fitness policies on children’s health promotion is negligible. It implies that there may be other non-linear pathways of action for this relational process. This may be due to the content of the national fitness policy, which is geared more towards adults and is increasingly focused on middle and old age health issues and less on adolescent and child populations. In addition, increasing population density has resulted in fewer places for a large number of children to participate in physical activity per capita, which somewhat inhibiting child physical development and thus increasing child mortality. On the other hand, although Chinese society is aware of the important role of physical activity participation in children’s physical and mental development, the implementation of health promotion policies generally lacks a good social awareness and supportive environment. Especially a passive physical education atmosphere from schools, communities, and families cannot meet children’s increasing health development needs. Therefore, it is necessary to analyze the transmutation characteristics and existing dilemmas of children’s sports health promotion policies from different perspectives and uses different methods in subsequent studies to explore more efficient paths for children’s health. In addition, the practitioner variable does not play a facilitating role in the national fitness policies. This is due to the phenomenon of redundancy caused by the excessive number of sports practitioners, which may reduce administrative efficiency and thus weaken the positive promoting effect of the national fitness policies. However, the financial expenditures on public sports per capita reinforce the positive effect of national fitness policies in promoting adult health. Abundant material resources can provide a solid foundation for the venue facilities, technical guidance, and physical therapy and rehabilitation needed to enhance citizens’ physical health.

#### 4.4.2. Regional Heterogeneity Analysis

Heterogeneity in the impact of national fitness policies on national health is also reflected across regions. We first compare the differences in health outcomes across regions in terms of national fitness policy intensity. In terms of direct effects, developed regions had higher levels of adult health, while health outcomes in underdeveloped regions are not significant. This may be because the overall level of economic and social development in underdeveloped regions is relatively low, resulting in an inadequate supply of public services for national fitness. The specific manifestations are: first, the serious lack of sports facilities and fitness venues exposes the problems of insufficient total supply, uneven distribution of supply resources and unreasonable utilization of resources. Second, the lack of professional sports personnel makes it difficult to meet the public’s demand for learning special sports skills, which in turn inhibits their enthusiasm for sports participation. Third, due to the limited regional openness and low overall health education level in underdeveloped regions, public awareness of participation in physical fitness activities needs to be enhanced. 

Further, the inclusion of practitioners and financial expenditures exacerbates the regional variability in the health effects of national fitness policies. This study argues that although the public service system of national fitness in developed regions is more complete, there may still be a problem of redundant practitioners. The lack of reasonable human resource allocation increases manpower costs, which may result in wasted public sport resources, reduce the positive experience of public participation in physical fitness activities, and lead to limited benefits for policy recipients, which in turn affects adult health. In those regions where economic and social development is relatively lagging behind, public services for national fitness are ranked low in the weighting of government policy-making matters, so sports practitioners play a limited role in the process of national fitness policies influencing national health. It applies to the role of sports financial expenditure in the stage process of national fitness policy affecting national health. The state has introduced targeted special financial support to broaden the coverage of public services for national fitness, which to a certain extent has alleviated the contradiction of balanced supply and demand of diversified and multi-level public demand for physical fitness. However, for regions and cities with lower levels of economic development, government policy-makers and administrative departments are not sufficiently aware of the importance of public services for national fitness to promote national health and sustainable regional economic development, and there is a gap in the allocation of sports resources and other specific measures compared to developed regions. In addition, the rapid but extremely unbalanced regional economic development in China, as well as the lack of clear boundaries in social responsibilities between government, market and individual, have further caused the supply of national fitness resources in different regions to exhibit inequitable characteristics. Some of the less developed regions, especially the central and western regions and rural areas, still have some urgent problems in the provision of public services for national fitness.

### 4.5. Robust Analysis

In order to ensure the rationality of the model and ensure that the regression is not an accidental observation result of a sample estimation, the robustness test of this study is implemented by replacing dependent variables and independent variables respectively after completing the above tests and analyses. The results are shown in Table 6. First, annual policy intensity is used instead of cumulative policy intensity. The regression results show that the higher the policy intensity per year, the lower the adult mortality. The higher the policy intensity per year, the lower the child mortality. Although not significant, its direction is consistent with the original results for child mortality. The results of the regression analysis obtained after substitution show approximately the same results as above. Second, the health index is used to replace adult mortality and child mortality. Considering the size and demographic differences of the provinces, the health index also has strong explanatory power compared to adult and child mortality. The results of the regression analysis obtained after substitution are still consistent with the above.

## 5. Conclusions

This study empirically examines the effect of national fitness policy intensity on population mortality in developed and underdeveloped regions by analyzing panel data on 30 Chinese provinces from 2008–2017 using an individual fixed effects model. Given that government investment in public sports in terms of personnel and funding may have important effects on national health, we take culture, sports, and media practitioner per 10,000 population and financial expenditures on culture, sports, and media per capita as moderating variables to assess the nonlinear relationship between national fitness policy intensity and national health. Based on data fitting and model derivation, the following main findings were obtained: First, the intensity of national fitness policy had a significant negative effect on adult mortality and child mortality, with a stable long-term equilibrium relationship in developed regions and a non-significant effect in underdeveloped regions. Second, culture, sports, and media practitioner per 10,000 population negatively moderates the relationship between national fitness policy intensity and adult mortality, but does not have a significant effect on the relationship between national fitness policy intensity and child mortality. When culture, sports, and media practitioner per 10,000 population increases significantly, the negative effect of national fitness policy intensity on adult mortality will be significantly weakened. Third, financial expenditures on culture, sports, and media per capita can significantly positively affect the relationship between national fitness policy intensity and adult mortality, but not with child mortality. The negative effect of national fitness policy intensity on adult mortality is significantly enhanced when financial expenditures on culture, sports, and media per capita increases. Finally, the relationships between policy intensity, culture, sports, and media practitioner per 10,000 population, financial expenditures on culture, sports, and media per capita, and adult mortality in developed regions are consistent with the national analysis, whereas sports, and media practitioner per 10,000 population, and financial expenditures on culture, sports, and media per capita, in underdeveloped regions do not exert significant effects. We speculate to some extent that there may be other factors that reinforce the effect of policy intensity on adult mortality. It further suggests that health research on public sports policies is still in its infancy in China, and thus there is a huge research space.

## 6. Discussion

This study comprehensively examines the efficiency of national fitness policies and their impact on national health in 30 Chinese provinces, and provides guidance for optimizing the current human resource allocation and financial investment patterns of China’s national fitness public service system, as well as improving national physical fitness and health. These findings have important policy implications. First, the study shows that national fitness policy intensity can significantly reduce population mortality. Provincial governments need to provide great importance to the development of national fitness sports, especially in the economically underdeveloped areas, to increase the policy support and efficiency of national fitness public services according to the content needs of the national fitness programs in the territory. Second, provincial governments should focus on developing sports practitioners to meet the requirements of public sports in the new era, strictly implement and establish a long-term responsibility mechanism, and improve the management efficiency of sports practitioners. It is necessary to avoid the redundancy and confusion of practitioners, which may bring constraints to the implementation of national fitness policies or practical operations. Third, a stable growth mechanism of financial expenditures for national fitness public services can be established. It can provide sufficient financial guarantee for national fitness, and promote scientific, refined and rationalized financial expenditure management of China’s national fitness public service system. Finally, the network effect, organizational effect, and scale effect of sports expenditures on national fitness should be examined, based on the impact of financial expenditures on national fitness public services. In-depth interviews should be conducted to explore the possible influence mechanisms of national fitness policy intensity in underdeveloped regions, ensuring that the development of national fitness in underdeveloped regions can also have sufficient public service resources for national fitness.

This study has some limitations that need to be improved and optimized in future studies. First, this study examines the impact of national fitness policy intensity on population mortality, and focuses on the efficiency of national fitness public services. However, only the quantitative dimension of policies is analyzed, and other dimensions such as quality, price, and acceptance of national fitness programs are not addressed. In future studies, we need to deepen the exploration of relevant impact mechanisms. Second, in terms of research sample and data analysis, other methods such as field questionnaires, face-to-face interviews, focus group discussions and small-scale policy experiments, can be introduced in future studies to further verify and expand the feasibility of existing findings. Research data of 30 provinces can also be refined for inter-provincial comparison studies to explore the stability and dispersion of provincial governments’ national fitness policy efficiency. Finally, future research can explore the multi-subject utility of national fitness, further determine their transmission path and mechanism that affect public sports investment and national health, and refine indicators for public service investment in national fitness.

## Figures and Tables

**Figure 1 ijerph-19-09191-f001:**
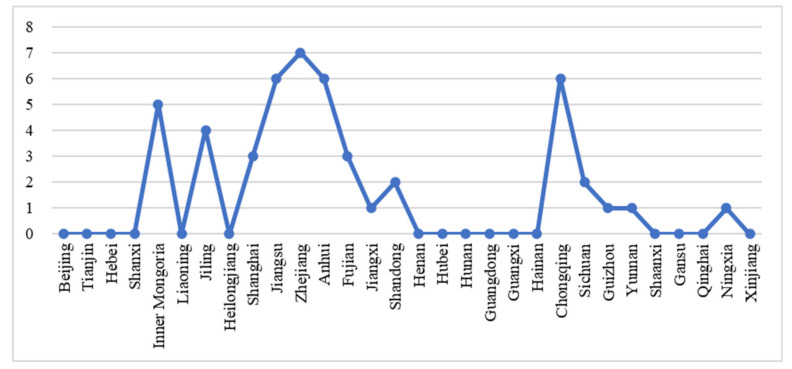
Initial intensity of national fitness policies in China in 2008.

**Figure 2 ijerph-19-09191-f002:**
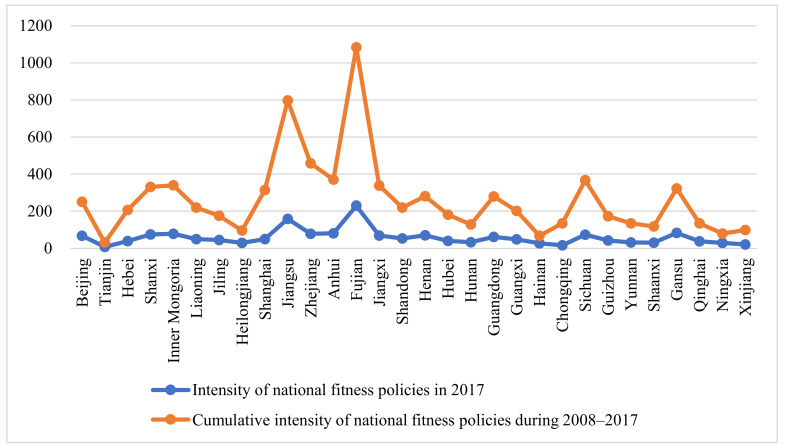
National fitness policy intensity and policy intensity accumulation of provinces in China in 2017.

**Table 1 ijerph-19-09191-t001:** Descriptive statistics.

Variables	Unit	Obs	Mean	SD	Min	Max
**Dependent variables**						
Policy intensity	Accumulating policy number	300	3.423224	1.811232	0.000000	6.989335
**Independent** **variables**						
Adult mortality	%	300	0.000092	0.000023	0.000038	0.000162
Child mortality	%	300	0.016183	0.007370	0.005000	0.043000
**Moderating** **variables**						
Culture, sports, and media practitioners per 10,000 population	person	300	2.367759	0.508242	1.524408	4.471974
Financial expenditures on culture, sports, and media per capita	yuan	300	5.048278	0.622657	3.682046	6.869663
**Control variables**						
Health care expenditures per capita	yuan	300	6.629206	0.497882	5.071565	7.901217
Health education institutes/stations	institute	300	1.385850	0.892431	0.000000	3.526361
PM_2.5_	ug/m^3^	300	3.551505	0.477682	2.186051	4.423648
Park green space per capita	m^2^	300	2.449364	0.233631	1.813195	2.984166
Urban sewage treatment rate	%	300	−0.189196	0.166759	−0.919045	−0.022246
Household waste harmless treatment rate	%	300	−0.191159	0.248369	−1.331049	0.000000
GDP per capita	yuan	300	10.541603	0.522224	9.195734	11.832077
Urbanization rate	%	300	−0.629852	0.229329	−1.234088	−0.109815
Population density	person per km^2^	300	7.843010	0.440769	6.475433	8.694000

Note: Logarithms are taken for the independent and control variables.

**Table 2 ijerph-19-09191-t002:** Regression results of equation.

	Adult Mortality			Child Mortality		
	Correlation Coefficient	SE	*p* > *t*	Correlation Coefficient	SE	*p* > *t*
Policy intensity	−0.000001	0.000000	0.009	0.000296	0.000105	0.005
Culture, sports, and media practitioners per 10,000 population	−0.000003	0.000003	0.176	0.001446	0.000651	**0.027**
Financial expenditures on culture, sports, and media per capita	−0.000003	0.000002	0.079	−0.000131	0.000428	0.760
Health care expenditures per capita	−0.000003	0.000002	0.112	−0.001719	0.000514	0.001
Health education institutes/stations	−0.000002	0.000001	0.098	−0.000363	0.000273	0.184
PM_2.5_	0.000000	0.000002	0.859	−0.000252	0.000466	0.590
Park green space per capita	−0.000009	0.000003	0.002	−0.003014	0.000703	0.000
Urban sewage treatment rate	−0.000004	0.000003	0.209	−0.003749	0.000727	0.000
Household waste harmless treatment rate	0.000001	0.000002	0.628	0.000148	0.000548	0.788
GDP per capita	0.000000	0.000003	0.948	−0.001182	0.000834	0.158
Urbanization rate	0.000011	0.000008	0.182	−0.017875	0.002115	0.000
Population density	−0.000003	0.000002	0.086	0.001627	0.000400	0.000

**Table 3 ijerph-19-09191-t003:** Regression results of equation.

	Adult Mortality			Child Mortality		
	Correlation Coefficient	SE	*p* > *t*	Correlation Coefficient	SE	*p* > *t*
Policy intensity	−0.000001	0.000000	0.004	0.000291	0.000118	0.014
Culture, sports, and media practitioners per 10,000 population	−0.000001	0.000003	0.791	0.001539	0.000688	0.026
Financial expenditures on culture, sports, and media per capita	−0.000004	0.000002	0.019	−0.000211	0.000433	0.627
Policy intensity * practitioners (interaction item)	0.000001	0.000000	0.005	0.000003	0.000105	0.977
Policy intensity * financial expenditures (interaction item)	−0.000001	0.000000	0.002	−0.000148	0.000097	0.127
Health care expenditures per capita	−0.000001	0.000002	0.806	−0.001238	0.000569	0.030
Health education institutes/stations	−0.000001	0.000001	0.194	−0.000342	0.000273	0.211
PM_2.5_	−0.000001	0.000002	0.616	−0.000512	0.000481	0.289
Park green space per capita	−0.000008	0.000003	0.003	−0.002650	0.000720	0.000
Urban sewage treatment rate	−0.000004	0.000003	0.158	−0.003801	0.000723	0.000
Household waste harmless treatment rate	0.000000	0.000002	0.994	−0.000193	0.000567	0.734
GDP per capita	0.000000	0.000003	0.889	−0.001278	0.000831	0.125
Urbanization rate	0.000010	0.000009	0.277	−0.020361	0.002468	0.000
Population density	−0.000002	0.000002	0.221	0.001713	0.000403	0.000

Note: Interaction items are policy intensity and practitioners, and policy intensity and financial expenditures. The table has been marked with “*”, the same below.

**Table 4 ijerph-19-09191-t004:** Regression results of equation.

	Adult Mortality in Developed Regions			Adult Mortality in Underdeveloped Regions		
	Correlation Coefficient	SE	*p* > *t*	Correlation Coefficient	SE	*p* > *t*
Policy intensity	−0.000001	0.000000	0.000	0.000000	0.000001	0.818
Culture, sports, and media practitioners per 10,000 population	−0.000007	0.000003	0.038	−0.000006	0.000003	0.094
Financial expenditures on culture, sports, and media per capita	0.000001	0.000001	0.374	−0.000008	0.000003	0.006
Health care expenditures per capita	0.000000	0.000002	0.929	−0.000003	0.000004	0.396
Health education institutes/stations	0.000001	0.000001	0.170	−0.000007	0.000002	0.001
PM_2.5_	0.000001	0.000002	0.486	0.000000	0.000003	0.934
Park green space per capita	−0.000008	0.000002	0.001	−0.000005	0.000005	0.290
Urban sewage treatment rate	0.000003	0.000004	0.443	−0.000007	0.000004	0.060
Household waste harmless treatment rate	0.000000	0.000002	0.916	−0.000001	0.000003	0.725
GDP per capita	−0.000002	0.000003	0.509	0.000000	0.000005	0.975
Urbanization rate	−0.000006	0.000007	0.405	0.000022	0.000017	0.201
Population density	−0.000009	0.000001	0.000	0.000005	0.000003	0.085

**Table 5 ijerph-19-09191-t005:** Regression results of equation.

	Adult Mortality in Developed Regions			Adult Mortality in Underdeveloped Regions		
	Correlation Coefficient	SE	*p* > *t*	Correlation Coefficient	SE	*p* > *t*
Policy intensity	−0.000002	0.000000	0.000	0.000000	0.000001	0.914
Culture, sports, and media practitioners per 10,000 population	−0.000003	0.000003	0.026	−0.000005	0.000004	0.019
Financial expenditures on culture, sports, and media per capita	0.000000	0.000001	0.088	−0.000008	0.000003	0.007
Policy intensity * practitioners	0.000001	0.000000	0.000	0.000000	0.000001	0.807
Policy intensity * financial expenditures	−0.000001	0.000000	0.000	0.000000	0.000001	0.804
Health care expenditures per capita	0.000003	0.000002	0.073	−0.000003	0.000004	0.464
Health education institutes/stations	0.000001	0.000001	0.111	−0.000007	0.000002	0.003
PM_2.5_	0.000000	0.000002	0.929	0.000000	0.000003	0.982
Park green space per capita	−0.000008	0.000002	0.001	−0.000005	0.000005	0.301
Urban sewage treatment rate	0.000002	0.000004	0.626	−0.000007	0.000004	0.062
Household waste harmless treatment rate	0.000000	0.000002	0.941	−0.000001	0.000003	0.694
GDP per capita	−0.000004	0.000003	0.212	0.000000	0.000005	0.996
Urbanization rate	−0.000001	0.000008	0.892	0.000022	0.000018	0.226
Population density	−0.000008	0.000001	0.000	0.000005	0.000003	0.085

**Table 6 ijerph-19-09191-t006:** Regression results of robustness analysis.

	Adult Mortality			Child Mortality			Health Index		
	Correlation Coefficient	SE	*p* > *t*	Correlation Coefficient	SE	*p* > *t*	Correlation Coefficient	SE	*p* > *t*
Policy intensity	−0.000001	0.000000	0.007	0.000151	0.000114	0.187	0.002423	0.000290	0.000
Culture, sports, and media practitioner per 10,000 population	−0.000003	0.000003	0.176	0.001481	0.000659	0.025	−0.003226	0.001799	0.074
Financial expenditures on culture, sports, and media per capita	−0.000003	0.000002	0.058	0.000080	0.000425	0.852	−0.000399	0.001181	0.736
Health care expenditure per capita	−0.000004	0.000002	0.067	−0.001502	0.000513	0.004	0.017375	0.001420	0.000
Health education institutes/stations	−0.000002	0.000001	0.083	−0.000356	0.000276	0.200	−0.000944	0.000753	0.211
PM_2.5_	0.000000	0.000002	0.803	−0.000167	0.000474	0.725	−0.005963	0.001288	0.000
Park green space per capita	−0.000009	0.000003	0.001	−0.002933	0.000710	0.000	0.003719	0.001941	0.056
Urban sewage treatment rate	−0.000003	0.000003	0.253	−0.003800	0.000737	0.000	−0.001509	0.002007	0.453
Household waste harmless treatment rate	0.000001	0.000002	0.623	0.000294	0.000554	0.596	−0.001017	0.001515	0.503
GDP per capita	−0.000001	0.000003	0.849	−0.000743	0.000827	0.370	0.013836	0.002304	0.000
Urbanization rate	0.000012	0.000008	0.137	−0.018021	0.002156	0.000	−0.022906	0.005843	0.000
Population density	−0.000003	0.000002	0.086	0.001619	0.000405	0.000	0.003401	0.001106	0.002

## Data Availability

Data used in this study are available from corresponding author, upon reasonable request.

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
