# Peer review of "Does the National Fitness Policy Promote National Health?—An Empirical Study from China"

_ijerph, 2022, doi:10.3390/ijerph19159191_

Round 1

Reviewer 1 Report

The authors present a study of the relationship between fitness policies and health outcomes. The main concern is that the outcomes are adult and child mortality rates. While adult mortality is somewhat relevant, child mortality is really a stretch. There is no clear definition of child age and it seems to not be a very useful metric since the fitness policy is not necessarily aimed at children. This is the major shortcoming of the study and needs to be clarified and justified better. Otherwise, the document is well-written with a few specific suggestions listed below:

Lines 27-31: Sentence is too long - consider separating.

Figure 1: This plot is on a scale of 0-8 which is not common. It would benefit the manuscript greatly if the authors could clarify the scale and what the values mean.

Figure 2: Similar to Figure 1 - what does the scale mean?

Line 290: Consider using a different word to not have "effect" twice in the same sentence.

Line 301: Please spell out "GBD" as is its first use.

Line 354 (and elsewhere): Subscript "2.5" in "PM2.5".

Line 374: What are the "two GDP per capita" indicators?

Table 1: It would be helpful to add units to the majority of the listed variables.

LInes 406-7: Please rewrite this sentence for clarity.

Line 416: p<0.1 is not statistically significant.

Table 3: What do the "*" mean in the first column?

Line 465: Instead of "were shown" use "are shown in Table 4".

Line 485: "P>0.1" should be "p>0.1".

Line 543: Has COVID-19 outcomes been regionally heterogeneous to justify this statement?

Line 593: Here it is written correctly "sport, and media". Elsewhere in the document the comma (",") is ommitted which is a mistake and should be corrected.

Line 629: Consider replacing "building sports practitioners" with "developing sports practitioners".

Reviewer 2 Report

With interest I have read this manuscript which is overall well-written, well-structured and clear.

Minor comments:

2.1. The Evolution of China’s National Fitness Policy in Practice

·        It should be explained why there are such large differences in the number of national fitness policies among different regions.

·        What types of policies were included in the study? Examples of such policies would be useful.

4.1. Direct Effect of National Fitness Policy Intensity on National Health

·        Line 415: There is a significant negative correlation between the national fitness policy intensity and adult mortality (r=-0.000001, p<0.1); the greater the national fitness  policy intensity, the lower the adult mortality rate. Was P value of less than 0.05 considered to be statistically significant?

Reviewer 3 Report

The overall aim of the paper is to examine the outcomes of the national fitness policy implemented following the Beijing summer Olympics in China and its provinces. The policies were examined as “national fitness policy intensity,” which was defined as the number of policies and regulations issued by leaders of Chinese provinces in a particular year or time- period. Using this variable, the authors of the paper reported the following which are the main contributions of the paper: 1) adult mortality decreased following implementation of the health policies, especially in more developed provinces; and 2) child mortality decreased in some provinces. For children the benefits of national fitness policy intensity in developed provinces may be masked by the fact that children appear to experience greater precarity in highly populated regions. While this possibility was briefly stated it was not adequately explored. Nonetheless, the strength of the paper is the robust analysis of how the national health policy developed by the Chinese government following the 2008 Olympics was taken up by its provinces and what effects it had on adult and child mortality.

General comments:

·       The paper is unfocused and as a result confusing, especially in earlier sections. However, in the last section (section 5) the first sentence clearly states the overall aim of the paper. This sentence explicitly states the overarching goal of the paper to examine the effect of national fitness policy intensity on mortality in developed and underdeveloped regions. It should be used to center the remainder of the paper.

·       The remainder of the paper is unfocused and confuses the reader with respect to whether the study is specifically examining fitness policies linked to public sports programs or all policies and regulations from 2008-2017. It becomes clear later in the paper that it’s all policies and regulations, including those linked to public sports, but the authors acknowledge that, “health research on public sports policies is still in its infancy in China.” Therefore, discussion of this aspect of the study should be significantly minimized and only introduced after the overarching aim and main contributions of the paper are robustly described.

·       The Introduction provides framing information about the role national fitness policies have played in many parts of the world. However, there is no information provided about Cuba where neighborhood-based fitness approaches have been used for decades to improve population health.

Specific comments:

  • Lines 324-325: The keywords used to comprehensively search for national fitness policies and regulations from 2008-2017 by province are missing. They should be provided to better understand how the “national fitness policy intensity” variable was determined.

Round 2

Reviewer 1 Report

The authors have improved the manuscript substantially and took into consideration suggestions. One suggestion that should be addressed is clarifying the scale of the first two figures. This was mentioned in comments 3 and 4 but not thoroughly addressed in the document.
